**Data Availability Statement:** All relevant data are within the paper and its Supporting Information files.

# Perceived learning difficulty associates with depressive symptoms and substance use among students of higher educational institutions in North Western Ethiopia: A cross sectional study

**Tesera Bitew**[1,2]*, **Wohabie Birhan**[1], **Demeke Wolie**[3]

1 Department of Psychology, Institute of Education and Behavioral Sciences, Debre Markos University, Debre Markos, Ethiopia, 2 Department of Psychiatry, African Mental Health Research Initiative (AMARI), Addis Ababa University, Addis Ababa, Ethiopia, 3 Department of Psychology, College of Education and Behavioral Sciences, Bahir Dar University, Bahir Dar, Ethiopia

* tesera2016@gmail.com

## Abstract

### Background

The potential role of perceived learning difficulty on depressive symptoms and substance use in the context of student population was seldom studied. This study aimed to investigate the association of perceived learning difficulty with depressive symptoms and substance use among university students in northwest Ethiopia.

### Methods

A cross sectional study was conducted on 710 pre-engineering students. A locally validated version of Patient Health Questionnaire (PHQ-9) was used to assess depressive symptoms at a cut off 5–9 indicating mild depressive symptoms and at a cut off 10 for major depressive symptoms. Perceived difficulty in school work was assessed by items dealing about difficulties in areas of course work. The response alternatives of these items were 0 = not at all, 1 = not so much, 2 = quite much, 3 = very much. The types of substances that students had used in their life and in the last three months were assessed. Negative binomial regression and multinomial regressions were employed to investigate the predictors of number of substance use and depressive symptoms respectively.

### Results

The prevalence of depressive symptoms was 71.4% (Mild: 30% and Major 41.4%). About 24.6% of participants had the experience of using at least one substance. Increment in perceived difficulties in learning score was associated with more use of substances (aRRR = 1.03, 95% CI: 1.01–1.06), mild level depressive symptoms (aOR = 1.10, 95% CI: 1.04, 1.56 and major depressive symptoms (aOR = 1.19, 95% CI: 1.13, 1.26). Every increment in anxiety score was associated with increased risk of mild level of depressive symptoms

**Funding:** This research was funded by Debre Markos University, 2016, Dr Tesera Bitew. TB is also financially supported by the Africa Mental Health Research Initiative (AMARI) within the DELTAS Africa Initiative [DEL-15-01] as a post-doctoral fellow. The DELTAS Africa Initiative is an independent funding scheme of the African Academy of Sciences (AAS)'s Alliance for Accelerating Excellence in Science in Africa (AESA) and supported by the New Partnership for Africa's Development Planning and Coordinating Agency (NEPAD Agency) with funding from the Wellcome Trust [DEL-15-01] and the UK government. But, the views in this study were fully that of the authors and were not that of the funder.

**Competing interests:** The authors have declared that no competing interests exist.

(aOR = 1.09, 95% CI: 1.01, 1.17) and major depressive symptoms (aOR = 1.28, 95% CI: 1.18, 1.37). Being male (aRRR = 5.54, 95% CI: 3.28, 9.36), urban residence (aRRR = 2.46, 95% CI: 1.62, 3.72) and increment in number of life threatening events (aRRR = 1.143, 95% CI: 1.08, 1.22) were associated with increased risk of substance use.

## Conclusion

Perceived difficulties in learning independently predicted increased depressive symptoms as well as substance use among participants.

## Background of the study

Depression during adolescence is a serious public health challenge which is associated with suicide [1, 2] reduced memory, lack of concentration and poor planning [3, 4], sadness, loss of interest on common task, hopelessness and lack of sleep and appetite [5]. Depression adversely affects adolescents' school and physical development [6]. Moreover, it is strongly associated with antisocial behaviours [7], substance use and anxiety [2] and poor social functioning [6]. Depressive disorders accounted for 40.5% of Disability Adjusted Life Years (DALYs) caused by mental and substance use disorders [8] posing high challenges especially on individuals at productive stage [9, 10]. Its estimated cost of lost productivity is 210 billion dollars per year in [11]. In Ethiopia, depression contributes about 6.5% of burden of diseases, which is the highest share of burden compared to other forms of mental disorders [12]. Its impact is even greater than the burden of disease contributed by Human Immuno-Virus (HIV), Tuberculosis (TB), or malaria.

The impact of depression among students of higher educational institutions may even be worse. Higher education students who are depressed may not be able to cope up with the academic and social activities. For example, previous studies depicted that children and adolescents with major depressive disorder performed less than those without depression in various cognitive domains [4]. University students who are depressed and unable to concentrate on their academics may ultimately drop out from the program or be dismissed.

Though depression is least diagnosed and least recognized among adolescents [13], it is a very common health problem among students in higher institutions in all parts of the world [14]. The prevalence of adolescent depression was 13.05% and 43.5% using Composite International Diagnostic Interview (CIDI) and the Beck Depression Inventory (BDI) respectively in a systematic review study [15]. The prevalence of adolescent depression in high income countries ranged from 17% in USA [16] according to the Patient Health Questionnaire–9 to 18.4% among females and 11.5% among males in Finland students [17] based on the Finnish version of BDI (Beck Depression Inventory). The prevalence was even higher among adolescents in Low and Middle Income countries (LMICs) ranging from 22% in India [18] to 36% in Kenya [19] at a Patient Health Questionnaire (PHQ-9) cut off 10.

In Ethiopia, the weighed prevalence of depression for the five studies, which had used Composite International Diagnostic Interview (CIDI), was 6.8% [20]. The Prevalence of depression among higher educational institutions of Ethiopia was about 41% of students in University of Gondar [21] using Symptom Reporting Questionnaire-20 (SRQ-20) at cut off 8; 32% in Addis Ababa University as assessed by Centre for Epidemiologic Studies- Depression Scale (CES-D) [22] and 32.2% in Ambo University using CESD [1].

Substance use among Ethiopian higher education institutions is considerably rising. For instance, a study in Axum University showed that 28.7 of students have lifetime khat chewing; 34.5% have been drinking alcohol and 9.5% of them smoking cigarette. Citing previous studies, these authors also mentioned that prevalence of substance use among students in Debre Markos town to be 14.1%. Likewise, a study in Addis Ababa University [2] showed that prevalence of drinking alcohol, chewing Khat and smoking among students was 31.4%, 14.1% and 8.7% respectively.

While depression and substance abuse are strongly associated [2, 23], they have also common risk factors. For example, poor academic performance [24], extreme poverty [25, 26] and social determinants such as school and family environment [24, 26] were common risk factors for both. Nevertheless, the potential role of perceived learning difficulty on depressive symptoms and substance use in the context of student population was not examined for future design of intervention strategies. We hypothesized that increased perceived learning difficulty would associate with increased depressive symptoms and increased use of substances controlling other potential confounders: anxiety, self-efficacy, social support, list of threatening events and academic performance. Thus, this study aimed to investigate the association of perceived learning difficulty with depressive symptoms and substance use controlling for the potential confounders.

## Methods

### Study setting

The study was conducted in three universities located in northwestern Ethiopia: namely Debre Markos University (DMU), Bahir Dar University (BDU) and Debre Tabor University (DTU). Both BDU and DTU are located around lake Tana where there is mass production of Khat through irrigation. Several sstudents who are mainly adolescents from different nations and nationalities of Ethiopia are placed in each academic year in these universities.

Though substance use was not a common practice of local people in the towns where these universities are located, some farmers have recently begun to produce khat for commercial purposes and this has increased its accessibility. As a result, chewing khat has become a common practice in these places especially, among students of higher institutions.

### Study design

A cross-sectional quantitative study was done among pre-engineering students of selected universities. Data about the students' socio-demographic variables, current status of depressive symptoms, substance use, anxiety, social support and experience of stressful life events were collected using survey questionnaire.

### Target population

Freshman students of 2016/2017 academic year were the target population. Each university had about 3000 freshman students within the given academic year. That is more than 9000 freshman students were attending their education from three selected universities. Eligibility criteria for selecting participants included being freshman student and not having disability hearing to the extent of impairing informed consent.

### Sampling techniques and sample size

We purposively selected Bahirdar University (BDU), Debre Markos University (DMU) and Debre Tabor University (DTU) all of which were found in northwestern Ethiopia. Each of

these universities represented first, second and third generation universities respectively. These universities were purposely selected based on accessibility and availability of substances in the universities' local areas. Among the student population, we purposely selected first year students in college of Technology for their high rate of attrition for another study. Thus, sample size was estimated for another study using single proportion formula with design effect of 1.5 and non-response rate of 10%. Accordingly, a total of 710 participants were required. Cluster sampling was used to recruit participants from the pre-engineering students (379 from DMU, 172 from DTU and 159 from BDU) where sections were units of clustering.

## Assessment

The outcome variable was depressive symptoms as assessed by using a locally validated version of PHQ-9. It has very good sensitivity and specificity to diagnose depressive symptoms and its severity forms at different cut offs: a score of five or more to indicate mild depressive symptoms and at a cut off 10 or more for major depressive symptoms [27]. Substance use was assessed by using a composite scale taken from previous study [22]. The ten items of the scale asked the life time and the three months experience of using five substances: khat, alcohol, tobacco, shisha and depressants [22]. The options were never (0), monthly (1), weekly (2) and daily (3). The instrument has internal consistency of 89%. A range of predictor variables were used in the study: Anxiety was assessed by Generalized Anxiety disorder scale (GAD) [25]; Perceived social support was assessed by OSSLO social support scale with three items [28]; Self-efficacy was assessed using a 10 item General Self-Efficacy Scale [29] and List of threatening events was assessed by a 12 dichotomous items asking whether students experienced the events in the last 12 months [21]. Perceived difficulty in school work was assessed by items taken from previous studies [17] that requests participants whether they had difficulties in areas of school work such as paying attention to teaching, teamwork, getting along with peers, getting along with teachers, doing homework, preparing for examinations, finding personal learning strategies, doing activities requiring initiative, doing reading tasks and doing writing tasks. The response alternatives were 0 = not at all, 1 = not so much, 2 = quite much, 3 = very much. Socio-demographic data such as participants' gender, age, residence and religion were collected. We compiled the data collection tools in a form of self-administered Amharic version of questionnaires which took an average of 20 minutes to complete.

## Data analysis

Statistical Packages for Social Sciences (SPSS) version 20 was used to conduct data analysis. Descriptive statistics was used to investigate prevalence of substance use and depressive symptoms. Negative binomial regression and multinomial regressions were employed to investigate the predictors of number of substance use and depressive symptoms respectively. Complete case analysis was used to treat missing data. STROBE Checklist [30] was used to ensure reporting of all relevant items for cross-sectional studies.

## Ethics approval and consent to participate

Ethical clearance was obtained from Ethics committee of Institute of Educational and Behavioral Sciences, Debre Markos University. Written consent was obtained from study participants and that was approved by the ethics committee. Participants were told about the objectives of the study and we informed them that they have the right to withdraw from the study or not to respond to any of the questions they didn't want to respond.

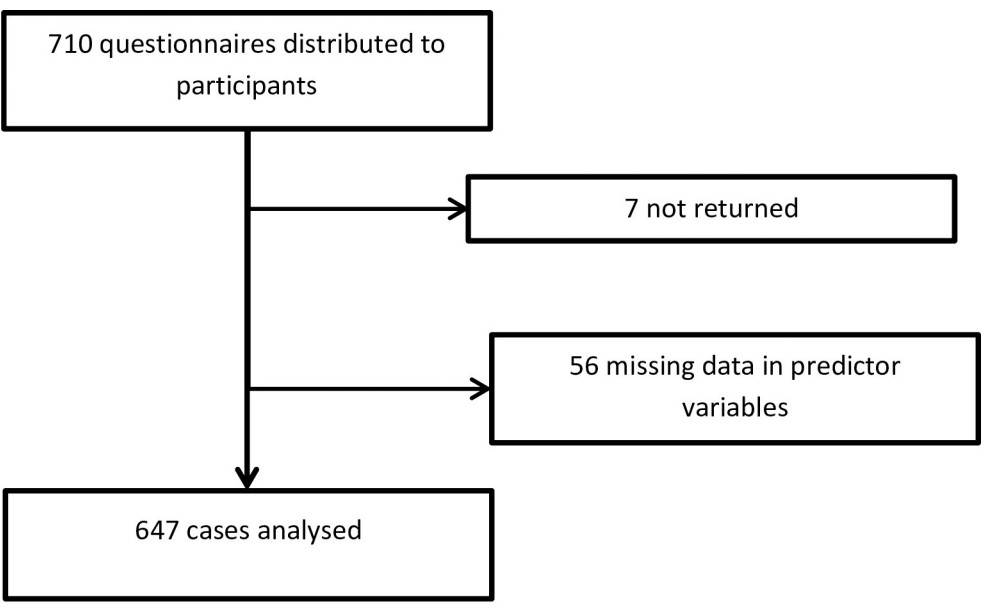

**Fig 1. Participant flow diagram.**

## Results

A total of 703 participants completed questionnaires out of 710 questionnaires with a response rate of 99%. But, 56 participants (9%) had missing data in either of the predictors and were excluded from analysis leaving 647 (91%) participants for multivariable regression analysis (Fig 1). The demographic background of the participants is indicated in Table 1. Mean age of participants was 19.91 years. More than two third of the participants were males; and most (60.5%) of them were residing in urban areas. The majorities (87.5%) of them were orthodox Christianity followers and about 6.4% of the participants had chronic illnesses.

As indicated in Table 2, the prevalence of mild level of depressive symptoms was about 30% and that of major depressive symptoms was about 41.4% which was statistically similar among both males and females (Mean PHQ-9 score = 9.09 and standard deviation = 6.58). Generally

**Table 1. Characteristics of the participants.**

| Characteristics | | Count | % |
|---|---|---|---|
| Sex | Male | 475 | 67.6 |
| | Female | 193 | 27.5 |
| Residence | Urban | 425 | 60.5 |
| | Rural | 267 | 38.0 |
| Religion | orthodox | 615 | 87.5 |
| | Muslim | 30 | 4.3 |
| | protestant | 35 | 5.0 |
| | Others | 9 | 1.3 |
| Chronic illness | Yes | 45 | 6.4 |
| | No | 653 | 92.9 |

Age (Mean = 19.91, minimum = 18, Maximum = 37, standard deviation = 1.48)

PHQ-9 total score (Mean = 9.09; Maximum = 27; Minimum = 0; standard deviation = 6.58)

**Table 2. Prevalence of depressive symptoms.**

| Variables | | Nill | Mild | Major | At least Mild |
|---|---|---|---|---|---|
| Sex | Males | 140 (29.5%) | 141 (29.7%) | 194 (40.8%) | 335 (70.5%) |
| | Females | 52(26.9%) | 63 (32.6%) | 78 (40.4%) | 141 (74.1%) |
| Residence | Urban | 128 (30.1%) | 136 (32.0%) | 161 (37.9%) | 297 (69.9%) |
| | Rural | 69 (25.8%) | 73 (27.3%) | 125 (46.8%) | 198 (74.2%) |
| University | DMU | 124 (32.7%) | 98 (25.9%) | 157 (41.4%) | 255 (67.3%) |
| | DTU | 39 (22.7%) | 60 (34.9%) | 73 (42.4%) | 133 (77.3%) |
| | BDU | 38 (25.0%) | 53 (34.9%) | 61 (40.1%) | 114 (75.0%) |
| | **Total** | **201 (28.6%)** | **211 (30.0%)** | **291 (41.4%)** | **502 (71.4%)** |

DMU: Debre Markos University; DTU: Debre Tabor University; BDU: Bahir Dar University.

about 70% of participants had a PHQ-9 score of five or more indicating probable depressive symptoms. Loss of interest on common tasks was the most frequent symptom among the participants (mean score = 1.61) followed by getting weak (mean score = 1.22) and hopelessness (mean score = 1.21).

Table 3 shows the three months prevalence of substance use. About one-fourth (24.6%) and nearly one-third (29.6%) of the participants had experience of using at least any one of the substance listed in Table 3 during the last three months and in lifetime respectively. Among the different forms of substances, alcohol was the most commonly (23.6% and 27.5%) used substance during the last three months and in lifetime respectively.

In negative binomial regression (Table 4), each increment in perceived difficulties in learning score was associated with increased risk of using more and more numbers of substances (aRRR = 1.03, 95% CI: 1.01, 1.05). There was more than four times increased risk of experiencing increased number of substances among males compared to females (aRRR = 4.44, 95% CI: 2.99, 6.58). With respect to residence, there was more than two times increased risk of experiencing greater number of substances use among urban residents (aRRR = 2.21, 95% CI: 1.65, 3.00) compared to rural residents. On the other hand each increment in PHQ-9 score (aRRR = 1.03, 95% CI: 1.01, 1.06) and number of life threatening events (aRR = 1.14, 95% CI: 1.08, 1.22) were associated with and increased risk of experiencing greater number of substances.

In a multinomial regression model aimed to investigate predictors of depressive symptoms (Table 5), each increment in perceived difficulties in learning score was associated with increased risk of having mild depressive symptoms (aOR = 1.10, 95% CI: 1.04, 1.16) and major depressive symptoms (aROR = 1.19, 95% CI: 1.13, 1.26).

Every increment in anxiety score was associated with 8.8% increased odds of mild level of depressive symptoms (aOR = 1.09, 95% CI: 1.01, 1.17) and 28% increased odds of major

**Table 3. Last three months and lifetime prevalence of substance use.**

| | Three months | | Lifetime | |
|---|---|---|---|---|
| Substance | Never | At least once | Never | At least once |
| Khat use in last 3 months | 675 (96.8) | 22 (3.2) | 653 (94.0) | 42 (6.0) |
| Shisha- use in last 3 months | 681 (97.7) | 16 (2.3) | 673 (97.0) | 21 (3.0) |
| Alcohol- use in last 3 months | 531 (75.4) | 166 (23.6) | 498 (72.1) | 193 (27.5) |
| Tobacco- use in last 3 months | 672 (95.4) | 25 (3.6) | 666 (96.1) | 27 (3.9) |
| Depressant—use in last 3 months | 682 (97.8) | 15 (2.2) | 679 (97.7) | 16 (2.3) |
| Experience of using either of the substances | 525(75.4) | 172 (24.6) | 481 (70.4) | 199 (29.6) |

**Table 4. Predictors of increased number of substance use in last 3 months in negative binomial regression model.**

| Parameter | 95% Wald Confidence Interval for RRR | | |
|---|---|---|---|
| | **RRR** | **Lower** | **Upper** |
| Gender: Males | **4.435**\*\* | 2.990 | 6.577 |
| Females | 1 | 1 | 1 |
| Residence: Urban | **2.207**\*\* | 1.649 | 2.955 |
| Rural | 1 | 1 | 1 |
| University: DMU | .884 | 0.624 | 1.251 |
| DTU | 1.055 | 0.730 | 1.525 |
| BDU | 1 | 1 | 1 |
| Having chronic illness: Yes | **2.541**\*\* | 1.679 | 3.846 |
| No | 1 | 1 | 1 |
| Anxiety Score | .988 | .956 | 1.021 |
| Self-Efficacy score | **0.978**\* | 0.959 | 0.998 |
| Perceived difficulties in learning | **1.032**\*\* | 1.011 | 1.053 |
| Social support | 0.982 | 0.945 | 1.022 |
| PHQ-9 score | **1.033**\* | 1.010 | 1.057 |
| Life threatening events | **1.143**\*\* | 1.076 | 1.215 |

\*\* p-value less than 0.005

\* p-value less than 0.05.

DMU: Debre Markos University; DTU: Debre Tabor University; BDU: Bahir Dar University.

depressive symptoms (aOR = 1.28, 95% CI: 1.18, 1.37). Each increment in number of substance used by students was also associated with about 55% increment in odds of major depressive symptoms (aOR = 1.55, 95% CI: 1.02, 2.36). Being urban resident and Debre Markos University student were associated with reduced odds of both mild (aOR = 0.40, 95% CI: 0.16, 0.70) and major levels of depressive symptoms (aOR = 0.34, 95% CI: 0.16, 0.70).

**Table 5. Predictors of depression in multinomial regression model.**

| Variables | Mild depressive symptoms | Major depressive symptoms |
|---|---|---|
| | **RRR (95%)** | **RRR** |
| Gender: Male | 0.735 (0.409, 1.324) | 0.745 (0.396, 1.405) |
| Female | 1 | 1 |
| Residence: Urban | **0.452**\* **(0.249, 0.820)** | **0.353**\*\* **(0.187, 0.664)** |
| Rural | 1 | 1 |
| University: DMU | **0.400**\***(0.165, 0.698)** | **0.340**\* **(0.165, 0.698)** |
| DTU | 0.827 (0.409, 2.135) | 0.934 (0.409, 2.135) |
| BDU | 1 | 1 |
| Anxiety score | **1.088**\* **(1.014, 1.168)** | **1.276**\*\* **(1.184, 1.374)** |
| Self-efficacy score | 1.008 (0.966, 1.053) | 0.967 (0.925, 1.010) |
| Perceived learning difficulties | **1.099**\*\***(1.044, 1.157)** | **1.191**\*\* **(1.129, 1.257)** |
| Social support score | 0.935 (0.854, 1.024) | 0.910 (0.826, 1.002) |
| Semester GPA | 0.774 (0.460, 1.301) | 0.682 (0.391, 1.190) |
| Number of substance | 1.341 (0.889, 2.022) | **1.551 (1.021, 2.355)** |
| Chronic illness: Yes | 0.545 | 1.248 (0.357, 4.365) |
| No | 1 | 1 |

DMU: Debre Markos University; DTU: Debre Tabor University; BDU: Bahir Dar University.

## Discussion

About 70% of participants had a PHQ-9 score of five or more indicating probable depressive symptoms (mild level of depressive symptoms is about 30% and that of 40% major depressive symptoms) which was statistically similar among both males and females. Loss of interest on common tasks, getting weak and sad mood were the most common depressive symptoms among the participants. Perceived learning difficulty was independently associated with increased risk of having depressive symptoms and increased risk of using different substances. This finding supports previous studies where low educational success was an antecedent to different risk behaviors including substance use as underlined in Social control theory [31]. This study has identified potential area of intervention, perceived difficulties in learning that can play a double role of improving both depressive symptoms and substance use at the same time. The positive association between perception of learning difficulties and use of substances may be explained by the students' use of psychoactive substances like khat for study purposes and the accessibility of khat in the study areas.

The prevalence of depressive symptoms varies across settings depending on cut offs and sample population. The prevalence of depressive symptoms in our samples is very high compared to the population study in southern Ethiopia which reported about 12% of depressive symptoms using the same measure [32]. But, the finding is comparable with a 43.5% prevalence of depressive symptoms using Beck depressive symptoms Inventory (BDI) in a Meta-analysis of about 53 international studies [15]. A 27.7% prevalence of depressive symptoms in Qaboos University using PHQ-9 diagnostic criteria at a cut point of more than 11 [33] was also comparable to our finding of 40% prevalence at cut off 10 using the same measure. Our finding supports a 41% prevalence of mental distress among students in University of Gondar [1, 21] and 32% prevalence of depressive symptoms in AAU as assessed by CEDS [22] and 32.2% in Ambo University using CESD [1].

Every increment in anxiety score and increased use of substances were associated with increased risk of having depressive symptoms. Anxiety arising from the challenges of adapting the new university environment among the rural freshman students may explain their increased risk of depressive symptoms and increased use of substances.

Alcohol is the most commonly used substance and nearly one fourth and one third of the participants used at least one of the substances in the last three months and in their lifetime respectively. In the same way, previous studies [34] in Europe found that substance use specially, alcohol, was the highest among adolescents. This is because of most easily accessibility of alcohol is the study area. Indeed, the current finding is somewhat similar to previous studies [35] which states that easy availability of substances increases use of substances. Likewise, a study conducted in Spain on adolescents [34] showed that there is high rate of substance use.

Previous studies, conducted on prevalence of substance use among students in Debre Markos town found it to be 14.1% [36]. Meanwhile research done in other higher institutions of Ethiopia showed that students are at risk of substance use. For instance, a study in Axum University showed that 28.7% of students have lifetime khat chewing; 34.5% have been drinking alcohol and 9.5% of them smoke cigarette.

The low prevalence of substance use in Debre Markos university can be explained by the relatively low availability of substances in the area and the cultural values of the community compared to Bahir Dar and Debre Tabor. In line with the effect of cultural, familial and environmental factors for substance use, empirical literature suggests that risk factors for drug abuse include having parents or siblings with problem of drug use, as well as family disruption, and poor attachment. The limitation of this study was that the time of data collection was nearer to final exam of the students. This might have inflated the prevalence of depressive

symptoms. The assessment tool we used for substances use was a composite measure and was not validated to assess the severity of the problem. Besides, a high rate of missing data in the predictors might have affected our results.

## Conclusion

Perceived difficulties in learning independently predicted increased depressive symptoms as well as substance use among adolescents.

## Supporting information

**S1 File.**
(SAV)

## Acknowledgments

We thank Dr Molalign Tamiru, Ato Temesgen Adam, Dr Askalemariam Adamu and Dr Demeke Binalf for their initiative to settle the inconvenience during the proposal development.

## Author Contributions

**Conceptualization:** Tesera Bitew.

**Data curation:** Tesera Bitew, Wohabie Birhan.

**Formal analysis:** Tesera Bitew.

**Funding acquisition:** Tesera Bitew, Wohabie Birhan.

**Investigation:** Tesera Bitew.

**Methodology:** Tesera Bitew.

**Project administration:** Tesera Bitew.

**Resources:** Tesera Bitew.

**Software:** Tesera Bitew.

**Supervision:** Tesera Bitew.

**Validation:** Tesera Bitew.

**Visualization:** Tesera Bitew, Wohabie Birhan.

**Writing – original draft:** Tesera Bitew.

**Writing – review & editing:** Tesera Bitew, Wohabie Birhan, Demeke Wolie.

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
