## [Decision Letter · Decision Letter 0]

20 Aug 2020

PONE-D-20-08001

Perceived learning difficulty associates with depressive symptoms and substance use among students of higher educational institutions in North Western Ethiopia: A cross sectional study

PLOS ONE

Dear Dr. Bitew,

Thank you for submitting your manuscript to PLOS ONE. After careful consideration, we feel that it has merit but does not fully meet PLOS ONE’s publication criteria as it currently stands. Therefore, we invite you to submit a revised version of the manuscript that addresses the points raised during the review process.

I could not get a comment from the Second reviewer unfortunately. But I think that the authors should address the methodological points the first reviewer pointed out.

We look forward to receiving your revised manuscript.

Kind regards,

Yutaka J. Matsuoka, MD, PhD

Academic Editor

PLOS ONE

Journal Requirements:

"This research was funded by Debre Markos University. But, the views in this study were fully

that of the authors and were not that of the funder.

TB is also financially supported by the Africa Mental Health Research Initiative (AMARI) within the

DELTAS Africa Initiative [DEL-15-01] as a post-doctoral fellow. The DELTAS Africa Initiative is an

independent funding scheme of the African Academy of Sciences (AAS)’s Alliance for Accelerating

Excellence in Science in Africa (AESA) and supported by the New Partnership for Africa’s Development

Planning and Coordinating Agency (NEPAD Agency) with funding from the Wellcome Trust [DEL-15-

01] and the UK government. TB was also supported financially by Debre Markos University."

"The funders had no role in study design, data collection and analysis, decision to

publish, or preparation of the manuscript."

4. Please ensure that you refer to Figure 1 in your text as, if accepted, production will need this reference to link the reader to the figure.

Reviewers' comments:

Reviewer's Responses to Questions

**Comments to the Author**

1. Is the manuscript technically sound, and do the data support the conclusions?

Reviewer #1: Partly

2. Has the statistical analysis been performed appropriately and rigorously? 

Reviewer #1: No

3. Have the authors made all data underlying the findings in their manuscript fully available?

Reviewer #1: No

4. Is the manuscript presented in an intelligible fashion and written in standard English?

Reviewer #1: No

5. Review Comments to the Author

Reviewer #1: The authors aimed to investigate the prevalence and risk factors of depressive symptoms and substance use among university students in North-Western Ethiopia. Focusing on depression and substance use is quite significant both from a clinical and public health perspective. However, several points should be critically considered.

[Major comments]

The authors assert the importance of clinical depression and substance abuse; however, this cross-sectional study assesses the level of depressive symptoms and substance use that compromise with a flaw, limiting the clinical relevance which needs to be considered.

The authors report that the prevalence of mild levels of depressive symptoms to be about 30%, and that of major depressive symptoms to be about 40%. However, the mean (SD) PHQ-9 score of the sample is not given, which makes it ambiguous whether the sample has marked depression symptoms that may cause noticeable problems in relationships with others and day-to-day school activities (functional impairment). The mean (SD) PHQ-9 score of the sample should be provided. The authors also report “mild” and “major” depressive symptoms based on PHQ-9; however, the score range of mild and major depressive symptoms category is not given.

The authors also report that substance use was assessed by a composite scale the asking the lifetime and the three months experience of using five substances: khat, alcohol, tobacco, shisha, and depressants, with an option of never (0), monthly (1), weekly (2) and daily (3). It appears that the scale does not evaluate noticeable problems (i.e., impairment) or distress that is an essential manifestation of substance use disorder; this assessment scale may show only recreational drinking or smoking. This makes it difficult to interpret the findings of this study whether it is referring recreational drinker or problematic alcohol users.

[Minor comments]

1) In Table 2, the prevalence of males having at least mild depressive symptoms is shown as 70.5% (n=235). According to my calculation it is 49.5% (n/N=235/475 = 0.4947), please check.

2) In Table 3, alcohol use in the last 3 months is shown as 173 (24.6%). The denominator should be all total substance users, 173/(28+25+173+33+24)=61.1%, please check.

3) English editing is needed.

6. PLOS authors have the option to publish the peer review history of their article (what does this mean?). If published, this will include your full peer review and any attached files.

Reviewer #1: No

---

## [Author Response · Author response to Decision Letter 0]

4 Sep 2020

Responses to reviewer comments

Response: thank you sending the links. We have now modified the manuscript (title page, author affiliations, the body of the manuscript and the file names accordingly. Since Plos One has its own section for funding, conflict of interest and availability of data, we have now removed the declarations section which contained these elements. 

Response: thank you. We have now uploaded the dataset as an additional file along with the revised manuscript. 

"This research was funded by Debre Markos University. But, the views in this study were fully

that of the authors and were not that of the funder.

TB is also financially supported by the Africa Mental Health Research Initiative (AMARI) within the

DELTAS Africa Initiative [DEL-15-01] as a post-doctoral fellow. The DELTAS Africa Initiative is an

independent funding scheme of the African Academy of Sciences (AAS)’s Alliance for Accelerating

Excellence in Science in Africa (AESA) and supported by the New Partnership for Africa’s Development

Planning and Coordinating Agency (NEPAD Agency) with funding from the Wellcome Trust [DEL-15-

01] and the UK government. TB was also supported financially by Debre Markos University."

"The funders had no role in study design, data collection and analysis, decision to

publish, or preparation of the manuscript."

Response: thank you indeed. We have now noted that our funding section submitted during our online submission stated as "The funders had no role in study design, data collection and analysis, decision to

publish, or preparation of the manuscript." We will update the funding information during submission. The update of the funding section will be made as follows. "This research was funded by Debre Markos University. But, the views in this study were fully

that of the authors and were not that of the funder.

TB is also financially supported by the Africa Mental Health Research Initiative (AMARI) within the

DELTAS Africa Initiative [DEL-15-01] as a post-doctoral fellow. The DELTAS Africa Initiative is an

independent funding scheme of the African Academy of Sciences (AAS)’s Alliance for Accelerating

Excellence in Science in Africa (AESA) and supported by the New Partnership for Africa’s Development

Planning and Coordinating Agency (NEPAD Agency) with funding from the Wellcome Trust [DEL-15-

01] and the UK government."

4. Please ensure that you refer to Figure 1 in your text as, if accepted, production will need this reference to link the reader to the figure.

 Response: thank you. We have no figure in this manuscript

Response: 

Reviewers' comments:

Reviewer's Responses to Questions

Comments to the Author

1. Is the manuscript technically sound, and do the data support the conclusions?

Reviewer #1: Partly

Response: we have now formatted the manuscript according the journal’s formatting guidelines. 

2. Has the statistical analysis been performed appropriately and rigorously?

Reviewer #1: No

Response: thanks for the comments of the reviewers on statistical aspects that guided as to revisit the statistical analysis again. In the current version of the manuscript, we have made modifications in Table 2, 3 and 4. We have checked the figures in the descriptive Table 2 and 3 using statistical package which was done using hand calculator. We have also added the descriptive for lifetime prevalence of substances in Table 2. 

3. Have the authors made all data underlying the findings in their manuscript fully available?

Reviewer #1: No

Response: we will be uploading the data accordingly. Thank you. 

4. Is the manuscript presented in an intelligible fashion and written in standard English?

Reviewer #1: No

Response: We have now carefully, edited the manuscript for language edits. 

5. Review Comments to the Author

Reviewer #1: The authors aimed to investigate the prevalence and risk factors of depressive symptoms and substance use among university students in North-Western Ethiopia. Focusing on depression and substance use is quite significant both from a clinical and public health perspective. However, several points should be critically considered.

[Major comments]

The authors assert the importance of clinical depression and substance abuse; however, this cross-sectional study assesses the level of depressive symptoms and substance use that compromise with a flaw, limiting the clinical relevance which needs to be considered.

Response: PHQ-9 is DSM based and very common tool in low income countries to assess depressive symptoms. It has been locally validated in Ethiopia with cut off of five and above indicating depressive symptoms. People with PHQ-9 score of five and above have functional impairment and above a cut off 10 or more is clinical level as suggested in validation studies. Indeed, we assume that the findings have clinical relevance taking the limitations into account. We have changed the substance use assessment scores in count data to reduce limitation of the substance use scale. Use of some of the substance like shisha and cannabis are very rare and illegal in Ethiopia. We also focused only on the last three month data than the lifetime to focus on severity though we inserted the lifetime prevalence for interested readers.

Moreover, We preferred the cross sectional survey as a first step just to know the prevalence of the problem. We plan to conduct clinical based research once we knew the extent of the problem. Hopefully, we will conduct clinical and intervention based design to help adolescent university students who are depressed due to substance use. 

The authors report that the prevalence of mild levels of depressive symptoms to be about 30%, and that of major depressive symptoms to be about 40%. However, the mean (SD) PHQ-9 score of the sample is not given, which makes it ambiguous whether the sample has marked depression symptoms that may cause noticeable problems in relationships with others and day-to-day school activities (functional impairment). The mean (SD) PHQ-9 score of the sample should be provided. The authors also report “mild” and “major” depressive symptoms based on PHQ-9; however, the score range of mild and major depressive symptoms category is not given.

Response: thank you we have now provided the mean and standard deviation of the PHQ-9 score at the bottom row of in Table 1 while interpreting it in the text on page 11 of the manuscript. The cut off score was reported in the abstract section of the manuscript. in the current version, we have also included in the main body under “assessment” section (page 9).*-------

The authors also report that substance use was assessed by a composite scale the asking the lifetime and the three months experience of using five substances: khat, alcohol, tobacco, shisha, and depressants, with an option of never (0), monthly (1), weekly (2) and daily (3). It appears that the scale does not evaluate noticeable problems (i.e., impairment) or distress that is an essential manifestation of substance use disorder; this assessment scale may show only recreational drinking or smoking. This makes it difficult to interpret the findings of this study whether it is referring recreational drinker or problematic alcohol users.

Response: We have changed the substance use assessment scores into count data to reduce limitation of the substance use scale. Use of some of the substance like shisa and cannabis are very rare and almost deviant behavior in Ethiopia. We also focused only on the last three month data than the lifetime to focus on severity though we inserted the lifetime prevalence for interested readers.

We have tried to check whether the respondents showed depressive symptoms or not due to substance use. Though not to the level of impairment, students reported that they used substances to escape from worries related to academic pressure , stress which may ultimately lead them to be dependent on these substance and unable to function without them. 

[Minor comments]

1) In Table 2, the prevalence of males having at least mild depressive symptoms is shown as 70.5% (n=235). According to my calculation it is 49.5% (n/N=235/475 = 0.4947), please check.

Response: Check. Yes, the last column was calculated by hand from the remaining cells obtained from SPSS cross tabulation. So, that was typo error. Thank you

2) In Table 3, alcohol use in the last 3 months is shown as 173 (24.6%). The denominator should be all total substance users, 173/(28+25+173+33+24)=61.1%, please check.

Response: the figure is referring to the prevalence and the denominator is the sample size. We have now made modifications and checked the figures for table 3.

3) English editing is needed.

Response: we have made careful language edits to the manuscript

6. PLOS authors have the option to publish the peer review history of their article (what does this mean?). If published, this will include your full peer review and any attached files.

Do you want your identity to be public for this peer review? For information about this choice, including consent withdrawal, please see our Privacy Policy.

Reviewer #1: No

---

## [Decision Letter · Decision Letter 1]

6 Oct 2020

Perceived learning difficulty associates with depressive symptoms and substance use among students of higher educational institutions in North Western Ethiopia: A cross sectional study

PONE-D-20-08001R1

Dear Dr. Bitew,

We’re pleased to inform you that your manuscript has been judged scientifically suitable for publication and will be formally accepted for publication once it meets all outstanding technical requirements.

Kind regards,

Yutaka J. Matsuoka, MD, PhD

Academic Editor

PLOS ONE

Reviewers' comments:

Reviewer's Responses to Questions

**Comments to the Author**

1. If the authors have adequately addressed your comments raised in a previous round of review and you feel that this manuscript is now acceptable for publication, you may indicate that here to bypass the “Comments to the Author” section, enter your conflict of interest statement in the “Confidential to Editor” section, and submit your "Accept" recommendation.

Reviewer #1: All comments have been addressed

2. Is the manuscript technically sound, and do the data support the conclusions?

Reviewer #1: Yes

3. Has the statistical analysis been performed appropriately and rigorously? 

Reviewer #1: Yes

4. Have the authors made all data underlying the findings in their manuscript fully available?

Reviewer #1: Yes

5. Is the manuscript presented in an intelligible fashion and written in standard English?

Reviewer #1: Yes

6. Review Comments to the Author

Reviewer #1: RE: PONE-D-20-08001 R1

The authors aimed to investigate the prevalence and risk factors of depressive symptoms and substance use among university students in North-Western Ethiopia. Points raised in the peer review were adequately addressed.

[Major comments]

The authors assert the importance of clinical depression and substance abuse; however, this cross-sectional study assesses the level of depressive symptoms and substance use that compromise with a flaw, limiting the clinical relevance which needs to be considered.

>>Authors state that they preferred the cross sectional survey as a first step just to know the prevalence of the problem. And have changed the substance use assessment scores into a count data to reduce limitation of the substance use scale, and focused only on the last three month data than the lifetime to focus on severity.

The authors report that the prevalence of mild levels of depressive symptoms to be about 30%, and that of major depressive symptoms to be about 40%. However, the mean (SD) PHQ-9 score of the sample is not given, which makes it ambiguous whether the sample has marked depression symptoms that may cause noticeable problems in relationships with others and day-to-day school activities (functional impairment). The mean (SD) PHQ-9 score of the sample should be provided. The authors also report “mild” and “major” depressive symptoms based on PHQ-9; however, the score range of mild and major depressive symptoms category is not given.

>>Authors have provided the mean and standard deviation of the PHQ-9 score in Table 1.

The authors also report that substance use was assessed by a composite scale the asking the lifetime and the three months experience of using five substances: khat, alcohol, tobacco, shisha, and depressants, with an option of never (0), monthly (1), weekly (2) and daily (3). It appears that the scale does not evaluate noticeable problems (i.e., impairment) or distress that is an essential manifestation of substance use disorder; this assessment scale may show only recreational drinking or smoking. This makes it difficult to interpret the findings of this study whether it is referring recreational drinker or problematic alcohol users.

>>Authors have changed the substance use assessment scores into a count data to reduce limitation of the substance use scale.

[Minor comments]

1) In Table 2, the prevalence of males having at least mild depressive symptoms is shown as 70.5% (n=235). According to my calculation it is 49.5% (n/N=235/475 = 0.4947), please check.

>>Table 2 has been revised.

2) In Table 3, alcohol use in the last 3 months is shown as 173 (24.6%). The denominator should be all total substance users, 173/(28+25+173+33+24)=61.1%, please check.

>>Table 3 has been revised.

7. PLOS authors have the option to publish the peer review history of their article (what does this mean?). If published, this will include your full peer review and any attached files.

Reviewer #1: No

---

## [Editor Report · Acceptance letter]

19 Oct 2020

PONE-D-20-08001R1 

Perceived learning difficulty associates with depressive symptoms and substance use among students of higher educational institutions in North Western Ethiopia: A cross sectional study 

Dear Dr. Bitew:

I'm pleased to inform you that your manuscript has been deemed suitable for publication in PLOS ONE. Congratulations! Your manuscript is now with our production department. 

Kind regards, 

on behalf of

Dr. Yutaka J. Matsuoka 

Academic Editor

PLOS ONE